# Androgens and NGF Mediate the Neurite-Outgrowth through Inactivation of RhoA

**DOI:** 10.3390/cells12030373

**Published:** 2023-01-19

**Authors:** Marzia Di Donato, Antonio Bilancio, Ferdinando Auricchio, Gabriella Castoria, Antimo Migliaccio

**Affiliations:** Department of Precision Medicine, University of Campania “L.Vanvitelli”, 80138 Naples, Italy

**Keywords:** androgens, nerve growth factor, small GTPases, neuronal commitment

## Abstract

Steroid hormones and growth factors control neuritogenesis through their cognate receptors under physiological and pathological conditions. We have already shown that nerve growth factor and androgens induce neurite outgrowth of PC12 cells through a reciprocal crosstalk between the NGF receptor, TrkA and the androgen receptor. Here, we report that androgens or NGF induce neuritogenesis in PC12 cells through inactivation of RhoA. Ectopic expression of the dominant negative RhoA N19 promotes, indeed, the neurite-elongation of unchallenged and androgen- or NGF-challenged PC12 cells and the increase in the expression levels of βIII tubulin, a specific neuronal marker. Pharmacological inhibition of the Ser/Thr kinase ROCK, an RhoA effector, induces neuritogenesis in unchallenged PC12 cells, and potentiates the effect of androgens and NGF, confirming the role of RhoA/ROCK axis in the neuritogenesis induced by androgen and NGF, through the phosphorylation of Akt. These findings suggest that therapies based on new selective androgen receptor modulators and/or RhoA/ROCK inhibitors might exert beneficial effects in the treatment of neuro-disorders, neurological diseases and ageing-related processes.

## 1. Introduction

In addition to controlling the reproductive functions, sex steroids exert a wide range of neuroprotective functions, sustain brain efficiency and influence mood. In this way, they attenuate the incidence of dementia [1]. Further, the protective role of sex steroids might be also related to steroid receptor expression after brain damage, as estrogens and androgens directly trigger reactive astrocytes and microglia, respectively [2]. These findings are of value, given the increasing interest about the role of glial cells in neuronal degeneration caused by the oxidative stress. Thus, steroid hormones act at different levels to prevent neuronal loss and limit brain damage.

Estrogens improve cognition through regulation of spine density, long-term potentiation and neurotransmitter systems. They also protect against neuron cell death and inhibit specific aspects of Alzheimer’s disease (AD), such as amyloid-β (Aβ) accumulation and tau hyperphosphorylation [3]. As such, an increased risk of AD is significantly affected by physiologic age-related depletion of the primary female sex steroids, estrogens and progestins [4]. Men also experience an age-related increase in the AD risk that involves a gradual decrease in bio-available testosterone, which typically progresses at a rate of ~2% per year in males. The decrease in male testosterone levels represents an important age-related risk factor for the neurodegeneration and AD development, a correlation that likely reflects the loss of beneficial androgen-mediated action in the brain [1,5]. 

Neuronal cells express the androgen receptor (AR) and androgens exert a neuroprotective role in the brain by inducing the axonal regeneration and stimulating neuron growth as well as synaptic function [6]. Androgens also protect against neuron cell loss and regulate AD-related pathology, including Aβ accumulation [7]. However, despite the pleiotropic effects exerted by androgens, a deep understanding of their action in the central nervous system (CNS) and brain functions is still pending. Clinical and experimental findings suggest, however, that restoring the androgen levels by hormone supply could be beneficial in the treatment of neurodegenerative diseases, mainly when serum testosterone levels are low [6,8]. Given these findings and the current absence of effective preventive and treatment interventions for neurologic diseases, there is an exciting interest in understanding the mechanism of the androgen/AR axis in the nervous system to pursue therapeutic strategies based on gonadal hormones [9].

One mechanism by which androgens may influence the nervous system’s functions is via aromatization to estradiol and the consequent activation of estradiol signaling, both genomic and non-genomic [10]. Nevertheless, our previous findings in PC12 cells and primary neurons have indicated that androgens act as endogenous regulators of neuronal functions. Genetic approach and siRNAs experiments have shown that a classic wild-type AR, rather than its splice variant(s), is involved in androgen neuronal functions [11]. By contrast, accumulation of abnormal AR is a pathogenic signature of spinal and bulbar muscular atrophy, a poly-glutamine-induced neurodegenerative disease [12] and AR splice variants have been detected in the nervous system [13,14,15]. Further, a 45 kDa AR splice variant (AR45), lacking the AR N-terminal domain, is expressed in several brain regions. Such a variant is localized to plasma membrane lipid rafts of dopaminergic neurons, suggesting that it may function as a membrane AR (mAR) to mediate fast, nongenomic androgen actions. It should be noted, however, that mAR, often insensitive to androgen antagonists, has not been cloned or sequenced so far [14,16] and the mechanism of action of mAR or AR variants often depends on the cellular environment analyzed [14].

We recently reported that nerve growth factor (NGF) and androgens induce neurite outgrowth of PC12 cells, through reciprocal crosstalk between the NGF receptor, TrkA, and classical AR. The actin binding protein, filamin A (FLNa) participates in this crosstalk through the involvement of β1 integrin. Once assembled, this complex leads to activation of the downstream PI3-K δ/Rac1 pathway [11]. Of note, FLNa is a versatile scaffold that regulates neuronal migration through rearrangement/stabilization of the actin cytoskeleton and interaction with extracellular matrix (ECM) as well as signal transduction molecules [17]. FLNa regulates neuronal progenitor proliferation and ensures a correct brain size through Wee-1 dependent Cdk1 phosphorylation [18]. Further, in periventricular nodular heterotopia, a congenital malformation commonly associated with FLNa mutations, postmitotic cortical neurons fail to migrate. At last, FLNa binds the small GTP binding proteins [19]. 

RhoA, Rac1 and Cdc42 belong to the Rho-family GTPases controlling a variety of processes, including neuronal morphogenesis, axon path-finding and dendritic spine formation/maintenance [20]. Changes in the activity and localization of GTPases and their downstream effectors account for cytoskeleton abnormalities and defects in dendritic spines as well as synapses observed in cognitive deficits, including AD [21,22]. The treatment of PC12 cells with NGF or androgens induces neurite elongation depending on Rac1 and Cdc42 activation [11]. However, in the aforementioned study, we failed to investigate the role of RhoA in the androgen- or NGF-modulated neuritogenesis. The present results, mainly obtained in PC12 cells and partially confirmed in hippocampal neurons, aim to fill this gap.

## 2. Materials and Methods

### 2.1. Chemical Reagents

The selective ROCK (p160ROCK) inhibitor, y27632 (Selleckchem, Houston, TX, USA), was used at 5 μM (final concentration). The TrkA inhibitor, GW441756 (Selleckhem), was used at 1 μM (final concentration). R1881 (Sigma Aldrich, Milan, Italy) was used at 10 nM. NGF (Millipore/Merck KGaA, Darmstadt, Germany) was used at 100 ng/mL. The antiandrogen bicalutamide (Sigma Aldrich) was used at 10 μM.

### 2.2. Cell Culture and Transfection

PC12 cells were kindly provided by Dr B. J. Eickholt (Charité–Universitätsmedizin Berlin, Cluster of Excellence NeuroCure, and Institute of Biochemistry–Charité CrossOver Ebene, Berlin, Germany). The cells were cultured using F12K medium (ATCC, Manassas, VA, USA) supplemented with 2.5% fetal calf serum (FCS; Life Technologies, Gaithersburg, MD, USA), 15% horse serum (Life Technologies), 100 μg/mL streptomycin (Life Technologies), and 100 U/mL penicillin (Life Technologies). Cells were made quiescent using phenol red–free DMEM (Life Technologies) containing 0.5% charcoal-treated FCS, 100 μg/mL streptomycin, 100 U/mL penicillin and 2 mM l-glutamine. They were transiently transfected with the dominant negative RhoA construct (RhoA N19; UMR Resource Center, Kansas, MO, USA), or the empty plasmid, pSG5, using the Superfect transfection reagent (Quiagen, Hilden, Germany). After 24 h, transfected cells were made quiescent for 24 h and then unstimulated or stimulated for the indicated times. 

### 2.3. Primary Mouse Hippocampal Neurons

C57/BL6 mice from our in-house colony were fed with a standard diet and water and maintained on a 12-h light/dark cycle. Primary cultures of hippocampal neurons were generated from gestational day 16.5 mice pups (*n* = 7–10 per preparation). Mouse embryos at 16.5 embryonic days (E16.5) were isolated and transferred into ice-cold DMEM. Hippocampi were dissected from E16.5 male and female mouse embryos [23] in phenol red high-glucose medium containing 10% horse serum, 2 mM l-glutamine, 100 U/mL penicillin and 100 U/mL streptomycin. Primary hippocampal neurons from both male and females were pooled and generated [23] using a cold Hank’s balanced saline solution (Life Technologies) and enzymatic dissociation (37 °C in 0.25% trypsin-EDTA (Lonza Bioscience, Siena, Italy) for 15 min and 10 mg/mL DNase I for 1 min). The resulting suspension was filtered through a 40-mm strainer (Corning, Corning, NY, USA) and used for a Western blot analysis or neuron’s cell culture. For neurite outgrowth assays, neurons were plated at a low density (50 cells/mm^2^) in Poly-D-Lysine (Thermofisher Scientific, Waltham, MA, USA) coated plates [11] and cultured in Neurocult Neuronal Plating medium (#05713; StemCell Technologies, Vancouver, Canada). After 4 h, when the cells were seeded, the medium was replaced with phenol-red free Neurobasal medium (Thermofisher Scientific) supplemented with 2% B27 (Thermofisher Scientific), 1 % Glutamax (Thermofisher Scientific), 100 U/mL penicillin and 100 U/mL streptomycin [24,25,26,27]. Animal procedures were done according to the Italian Legislative Decree 116/92 issued by the Italian Ministry of Health, as well as European Community laws.

### 2.4. Neurite Outgrowth Assay, Contrast Phase and Immunofluorescence (IF) Microscopy

To examine neurite outgrowth, the cells were plated at a density of 5 × 10^3^ cells/well in 60 mm Poly-D-Lysine (Thermofisher Scientific) coated plates, made quiescent and then left untreated or treated with the indicated compounds for 24 or 48 h. Cells were then analyzed by contrast phase microscopy and counted for neurite outgrowth. The number of neurite-bearing cells was determined by counting the number of cells in 30 arbitrary areas on the dish, each containing at least 30 single cells. A cell was identified as positive for neurite outgrowth if elongations were at least equal to or greater than two-fold the cell bodies in length. Cells were visualized by contrast phase microscopy with a DMIRB inverted microscope (Leica, Wetzlar, Germany) using N-Plan 10×, 20×, and 40× objectives (Leica). Images were captured using a DC200 or DFC 450C camera (Leica) and acquired with Application Suite (Leica) software. They are representative of at least three different experiments, each performed in duplicate.

Cytoskeleton analysis was done [28] using Texas red-labeled phalloidin (Sigma Aldrich). Neuronal class III β-tubulin was revealed using diluted (1:500 in PBS) Alexa Fluor–labeled neuronal class III β-tubulin (TUJ1) monoclonal antibody (Covance, Pinceton, NJ, USA) [11].

### 2.5. Lysates, RhoA Pull down Assay and Western Blot

For RhoA pull down assay, PC12 cells were collected, washed once with ice-cold PBS, and then lysed in lysis buffer following the manufacturer’s instructions. Lysates were centrifuged and supernatants, equalized for protein concentration, were incubated with glutathione-s-transferase (GST)-tagged fusion protein corresponding to residues 7–89 of mouse Rhotekin (Rhotekin-Rho binding domain (RBD) for GTP-RhoA measurement). Bound proteins were eluted, and samples were electrophoresed and analyzed by Western blotting with the mouse monoclonal anti-RhoA antibody (# 05-778; Millipore, Burlington, MA, USA).

Lysates were obtained as detailed in [29]. The following antibodies were used: mouse monoclonal anti-AR antibody (441; Santa Cruz Biotechnology, Dallas, TX, USA); rabbit polyclonal anti-TrkA antibody (06-574; Millipore); anti-phospho-Ser-473 Akt and anti-phospho-GSK-3α/β (Ser21/9) (#9271; #9272; #9331; Cell Signalling technology, Danvers, MA, USA) respectively [30]). α-Tubulin or glyceraldehyde-3-phosphate dehydrogenase (GAPDH) were detected using mouse monoclonal anti-tubulin or anti-GADPH antibodies (Sigma Aldrich). Neuronal class III β-tubulin was revealed using the mouse monoclonal anti-III β-tubulin (TUJ1) antibody (Covance, Pinceton, NJ). The ECL system (GE Healthcare, Chicago, IL, USA) was used to reveal immune-reactive proteins.

### 2.6. Statistical Analysis and Data Availability Statement

Experiments were performed in triplicate and data presented as the mean ± standard deviation. Where appropriate, two-tailed unpaired Student *t*-tests and one-way or two-way ANOVA (Bonferroni’s post-hoc test) were used. A *p* value < 0.05 was considered significant.

## 3. Results

### 3.1. Androgens and NGF Promote the Neurite-Outgrowth through the RhoA Inactivation

Rat adrenal pheochromocytoma PC12 cells [31] have been largely employed to study neuronal differentiation. These cells undergo differentiation when challenged with NGF, while there is proliferation on epidermal growth factor (EGF) stimulation [32]. In addition to expressing the NGF receptor, TrkA (Figure 1A), PC12 cells harbor low amounts of full-length AR, as shown by genetic and somatic knockdown approaches as well as Western blot analysis ([11] and Figure 1B). Cells challenged with androgen and NGF exhibit a modification of the cell shape and the formation of dendritic spines, as shown by the staining of F-actin [11] in Figure 1C. R1881 and NGF consistently promote the neurite elongation, as assessed by quantification from several experiments (Figure 1D). Because of the role of androgen/AR axis on Rac1 and Cdc42 activation in these cells, we here analyzed the androgen regulation of RhoA. The non-aromatizable androgen, R1881, inhibits within a few minutes RhoA activation, as shown by pull down assay in Figure 1D. The antiandrogen, bicalutamide and the specific TrkA inhibitor, GW441756 [33], both reverse this effect, indicating the involvement of AR and TrkA in androgen-induced RhoA inhibition. Superimposable results were observed on NGF treatment of the cells (Figure 1E). Bicalutamide and GW441756 both reverse the NGF effect, further corroborating the previously described cross talk between androgens and TrkA in various cell types, including PC12 cells [11,34]. 

### 3.2. Inactivation of RhoA Is Implicated in Androgen- and NG- Induced Neurite Outgrowth of PC12 Cells

Given the findings on hormone regulation of RhoA activity, we next assessed the effect of the dominant negative RhoA N19 [35] in the androgen signaling. Figure 2A shows that PC12 cells transfected with RhoA N19 construct actually overexpress the small GTP binding protein, as compared with cells transfected with the pSG5 empty vector. These cells were then used for the subsequent experiments. Irrespective of ligand stimulation, contrast-phase images in Figure 2B show that RhoA N19 overexpression promotes the neurite elongation in PC12 cells, as compared with the corresponding cells transfected with the control vector. R1881 or NGF slightly, but significantly, increases such an effect. Data from several independent experiments are quantified and presented in panel C.

βIII tubulin represents a marker of neuronal commitment in normal tissues [36,37]. We then analyzed the effect of RhoA N19 overexpression on modulation of this marker, as readout of neuronal commitment. The WB in Figure 2D shows that under basal conditions, the RhoA N19-transfected cells exhibit high βIII tubulin expression levels, as compared with cells transfected with control plasmid. Albeit at weak extent, overexpression of RhoA N19 significantly increases βIII tubulin expression in R1881- or NGF-stimulated cells. WBs from different experiments were quantified and graphically shown in Figure 2E. These findings strongly support the view that GDP-loaded RhoA is involved in the earliest phases of neuronal differentiation. 

### 3.3. The Inhibition of ROCK Enhances Neuritogenesis in Androgen- and NGF-Challanged PC12 Cells

Once activated, RhoA interacts with several effectors, including the Ser/Thr kinase ROCK [38,39] that is expressed in many types of nerve cells in the CNS [40,41]. In the contrast-phase images in Figure 3A and the IF images in Figure 3C, obtained by the immunostaining with the neuronal cytoskeleton antigen, the class III β-tubulin isotype shows that androgens or NGF promote the neurite outgrowth. 

We next used the y27632 ROCK inhibitor, which impairs the ROCK catalytic activity by competing with ATP for binding to the catalytic site [42]. y27632 treatment not only induces neurite-outgrowth in unchallenged PC12 cells, but significantly potentiates the effect of R1881 and NGF (Figure 3A,C), confirming the contribution of RhoA/ROCK axis in androgen- or NGF-induced neuritogenesis. Here again, data collected from several experiments in A were quantified and presented in B. The expression level of III β-tubulin isotype antigen was increased by R1881 or NGF, as assessed by WB in PC12 cell lysates and also in this case the ROCK inhibitor potentiates their effects (Figure 3D). 

Since regeneration of sensory axons usually involves the PI3K/Akt pathway and NGF-induced neurite outgrowth requires Akt [43], we finally assessed the role of GSK 3α/β/Akt activation in y27632-induced neurite outgrowth. Figure 3E,F show that R1881 (E) or NGF (F) induce within 5 min GSK 3α/β and Akt phosphorylation. Both the effects are robustly increased by treatment of PC12 cells with y27632. 

### 3.4. AR, TRKa and ROCK Are Involved in Neuritogenesis of Hippocampal Neurons in Primary Cultures

To strengthen our findings in PC12 cells, we next used primary mouse hippocampal neurons. Irrespective of sex, female and male-derived hippocampal neurons express both the AR and estrogen receptor [44]. In our experimental setting, these cells express appreciable amounts of AR and robust levels TrkA, as shown by the WB analysis presented in Figure 4A. Contrast-phase images in Figure 4B show that after 4 h the cells were adherent but had not yet a neuronal-like morphology. Here, we show neuronal cultures at 1 day in vitro (1DIV) and 3DIV, since these developmental stages correspond to neurite outgrowth and dendritic branching, respectively. Bicalutamide and GW441756 significantly reduced the elongation of primary neurites in hippocampal neurons at 1 DIV and the intersection of the neurites at 3 DIV. Notably, the ROCK inhibitor y27632 consistently enhances the neurite elongation in primary hippocampal cells, which form a typical neuronal network, already appreciable after 1 DIV of culture (Figure 4B). These results corroborate our findings in PC12 cells.

In summary, the present study, together with our previous findings [11], shows that androgen action in neuronal cells impinges on the simultaneous activation of Rac1 and inactivation of RhoA. The net result of hormonal control of this signaling pathway likely ensures a proper neuritogenesis (Figure 4C). 

## 4. Discussion

The role of androgen/AR axis in neuritogenesis and neurodegenerative diseases is still debated, with most studies performed in neuronal cells. Nevertheless, the interest of neurobiologists has been also focused on the androgen action in glial cells [2], because of the role of these cells in maintaining the brain environment [45], energy storage [46], synaptic maintenance [47] and synthesis of neurotrophic factors as well as neurosteroids [48,49]). 

Our previous evidence in quite different cell types, including primary neurons, PC12 cells [11] and prostate cancer-derived LNCaP cells [34] has shown a significant intersection between androgens and NGF in these cells. The link we discovered relies on a bidirectional cross talk between the NGF receptor, TrkA and AR. Such interaction controls neuritogenesis in PC12 cells or mitogenesis and invasion in LNCaP cells. The findings we reported in PC12 cells were exclusively obtained by avoiding ectopic expression of full-length AR, as PC12 cells harbor low amounts of endogenous AR, which is localized at the extra-nuclear level and is devoid of transcriptional activity. Moreover, R1881- or DHT-ligand bound AR mediates the responses we previously described [11]. Although previous studies have reported that dehydroepiandrosterone (DHEA) inhibits the NGF-induced MAPK activation [50] or triggers the survival [51] in PC12 cells, we did not investigate the effect of DHEA in our experimental setting, as the exact nature of the receptor systems mediating the rapid and potent actions of DHEA in PC12 cells still remains unknown. 

The translational impact of androgen or NGF-triggered AR/TrkA complex assembly we detected in PC12 cells (and other target tissues) [34,52] is evident, since androgens or NGF may substitute each other’s in sustaining neuronal differentiation or transformed features of target cells. As such, TrkA might undergo activation upon a local increase in androgen levels, which frequently occurs in mammalian brain [53]. In this regard, it should be mentioned that mammalian brain expresses both the isoforms of the enzyme 5α-reductase, type 1 (5α-R1) and type 2 (5α-R2), which convert testosterone to a more potent androgen, dihydrotestosterone (DHT) that participates in the sexual differentiation processes of some brain regions [54,55,56,57,58]. In such a way, a local increase of DHT might be responsible for TrkA activation on one hand. On the other, NGF might activate AR. As a footnote on the synergism between steroid endocrine system and neurotrophins, many studies have previously pointed to this aspect [59,60,61,62,63,64,65]. Perturbation of this balance would enable the NGF signaling derangement in neuronal cells that express a *plethora* of steroid receptors. 

As before introduced, FLNa participates in AR/TrkA crosstalk through the involvement of β1 integrin. Once assembled, this complex fosters activation of the Cdc42/Rac1 downstream pathway leading to cytoskeleton changes and neuritogenesis [11]. The present data show that androgens or NGF simultaneously impair the signaling leading to RhoA activation, and they both decrease the amount of GTP-loaded RhoA. Such effect is likely caused by FLNa recruitment to AR/TrkA complex, since FLNa intersects the small GTP-binding proteins through its action on Trio-GEF as well as FLN-GAP [19]. As FLNa does not discriminate between the GTP or GDP loaded forms [66], it might be argued that FLNa modifies the intracellular localization of the small GTP binding proteins, allowing the activation of Cdc42/Rac1 on one hand. On the other, FLNa might restrain the signalling of RhoA. The net balance between activation/inactivation of small GTPases by androgens or NGF likely ensures proper cytoskeleton changes allowing the neuritogenesis. 

Rapid steroid activation of small GTP-binding proteins is not unexpected and has been so far observed in different cell types [28,29,67,68,69,70,71,72]. Nevertheless, the RhoA inhibitory effect by androgens or NGF we report is novel. Experiments with the dominant negative version of RhoA, RhoA N19, reinforce these findings. Further, we observe that ROCK, a downstream effector of RhoA, is strictly implied in this process. ROCK is expressed in CNS, where its hyper-activity may lead to oxidative stress, uncontrolled inflammation, immune abnormality, energy metabolism disorders, neuronal cell loss, reactive gliosis, and impaired synaptic transmission. By this way, ROCK promotes the development of neurodegenerative diseases [73]. Its overexpression has been detected in the lesions of AD, Parkinson’s disease and multiple sclerosis [74,75,76]. As such, inhibition of ROCK activity is considered a promising strategy for neuro-regenerative approaches. Moreover, pharmacological inhibition of ROCK by the small molecule y27632 has been already demonstrated effective in axonal regeneration after traumatic lesions [77]. Although we did not assay ROCK activity, experiments with y27632 inhibitor confirm that ROCK catalytic activity is required for the androgen- or NGF-induced neuronal commitment of PC12 cells. They also show that activation of both Akt and GSK IIIβ is involved in this control. The findings obtained in primary mouse hippocampal neurons treated with y27632 further corroborate the role of ROCK in the observed effects. 

ROCK inhibitors have shown a great efficacy in preclinical models of neurologic diseases [78] and they are currently employed in clinical trials in patients with tauopathies (NCT04734379). By contrast, long-term studies with androgen supply in a large cohort of patients are required to draw any conclusion on the effect of androgens on neurodegenerative diseases. However, since in our experimental setting we did not evaluate the synergistic or additive effects of ROCK inhibitor and AR agonists, further studies would clarify the benefit of combinatorial therapy over the monotherapy. 

We are aware that our findings prevalently derive from cultured cells. Nonetheless, PC12 cells are still preferred as model in neurobiology studies because of their characteristics after the differentiation, such as neurotransmitters secretion, neuron morphology as well as ion and neurotransmitter receptors [79]. Being extensively used, PC12 cells come with the advantage of a great amount of knowledge about culturing conditions and differentiation process. However, the findings we observe in PC12 cells have been partially reproduced in primary hippocampal neurons. 

Hippocampal neurons express the same level of aromatase and 5α-R irrespective of sex [80,81], cultivated hippocampal neurons derived from male or female animals synthesize similar amounts [81] of estrogens [82,83] or DHT [84] and express both ERs and ARs [85]. However, the role of embryo sex in our experimental setting cannot be excluded. Indeed, while the capability of estradiol synthesis is independent of gender in vitro [82], the content of hippocampal estradiol is higher in female then in male rats in vivo [83]. Again, estrous cyclicity sustains the hippocampal neurons spine number, while gonadectomy reduces it in both female and male mice [86,87]. Additionally, experimental findings have argued for a pivotal role of brain-derived sex steroids in synaptic plasticity [88,89]. However, although spine synapse loss after gonadectomy leads to sex differences in synaptic plasticity, evidence to decipher the precise roles of estrogen and testosterone/DHT did not focus on sex differences in the steroid responsiveness [82,83,85,90,91,92].

In sum, it is conceivable that every cell in the male brain might differ from those in females, because to the differences in their sex chromosome complement or in response to the hormonal effects, by keeping in mind that hormone levels may vary over the cell lifespan and are highly influenced by exogenous supply or endogenous fluctuations [93]. Thus, there are still in vivo/in vitro discrepancies, although the recent discoveries in vivo suggest that the synthesis of neurosteroids is regulated in a sex-specific manner in the hippocampus. In conclusion, the sex needs to be considered in neurosciences, although differences between sexes might prevalently emerge if the studies are performed in vivo [85,91]. 

Further experiments in mouse models are required to dissect the sex-specific responsiveness to steroids as well as the non-genomic AR actions in hippocampal neurons. 

## Figures and Tables

**Figure 1 cells-12-00373-f001:**
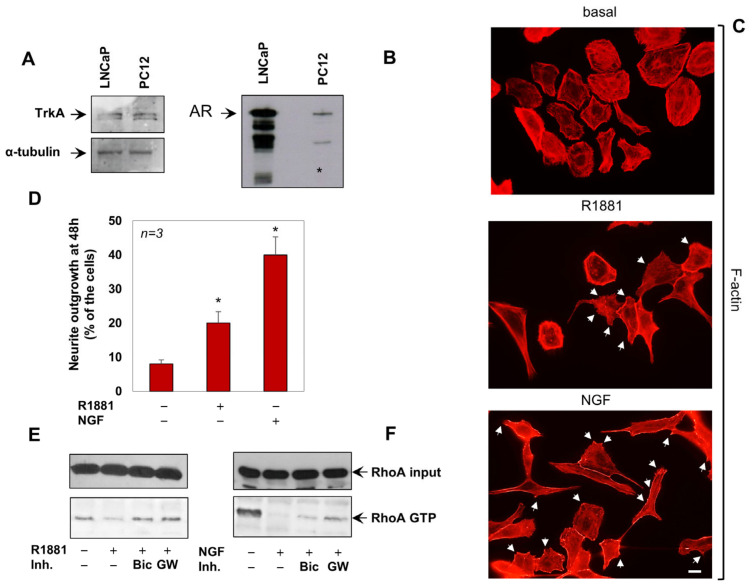
NGF and R1881 induce the RhoA inactivation and neurite outgrowth in PC12 cells. In (**A**,**B**), lysates proteins prepared from the indicated cell lines were analyzed by WB, using the antibodies against the indicated proteins. (**C**), PC12 cells were untreated or treated for 24 h with R1881 or NGF, stained for F-actin using Texas red–phalloidin and analyzed by IF. Images are representative of three independent experiments. Arrows indicate the dendritic spines. Bar, 10 μM. (**D**), Neurite outgrowth was analyzed as reported in Methods. Data derived from three different experiments were analyzed and expressed as percentage of total cells. Means and SEM are shown; *n* represents the number of experiments; * *p* < 0.05. (**E**,**F**), cells were left untreated or treated for 5 min with R1881 (left) or NGF (right), in the absence or presence of Bicalutamide (Bic) and GW441756 (GW). The GTP-loaded form of RhoA (RhoA-GTP) was assayed as in Methods and revealed by WB. Loaded RhoA (RhoA input) was also detected, as control.

**Figure 2 cells-12-00373-f002:**
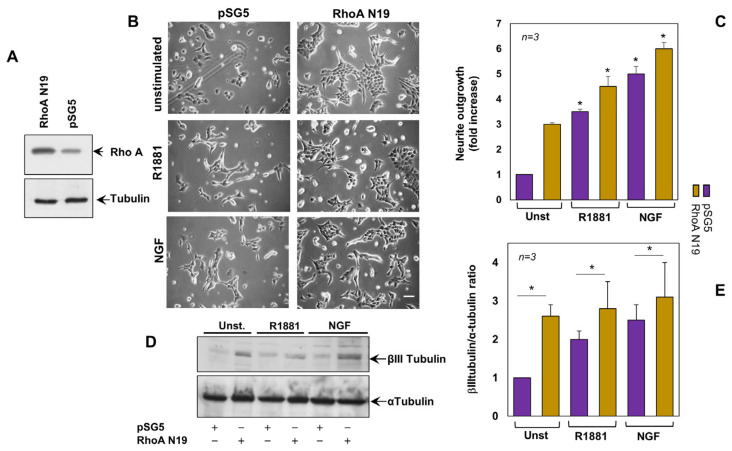
Expression of the dominant negative RhoA (RhoA N19) induces neurite outgrowth. (**A**), PC12 cells, transiently transfected with RhoA N19 construct or the empty plasmid pSG5, were untreated or treated for 24 h with R1881 or NGF and neurite outgrowth was analyzed by contrast phase microscopy (**B**). Bar, 10 µM. In (**C**), the neurite outgrowth was expressed as fold increase over the basal level, which corresponds to unstimulated cells transfected with pSG5. It was significant (* *p* < 0.05) for the indicated experimental points vs the corresponding unstimulated cells. Lysate proteins from transfected cells were analyzed for βIII or α tubulin expression (**D**). Protein amounts in (**D**) were quantified by NIH Image J software and graphically shown as βIII/α tubulin ratio (**E**). * *p* < 0.05. In (**C**,**E**), means and SEMs are shown. *n* indicates the number of experiments.

**Figure 3 cells-12-00373-f003:**
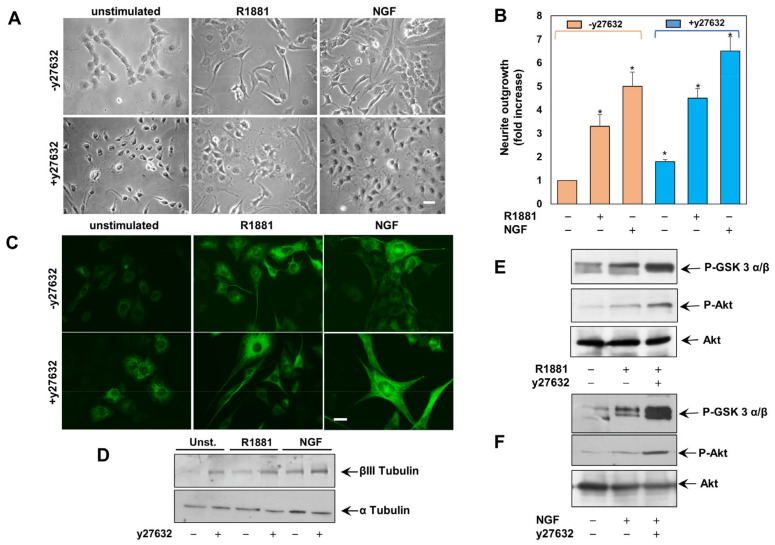
ROCK inhibition increases the R1881- and NGF-neurite elongation. (**A**), PC12 cells were untreated or treated for 24 h with R1881 or NGF in absence or presence of y27632 and neurite outgrowth was analyzed (**B**). (**B**), the difference in neurite outgrowth was expressed as fold increase over the basal levels. * *p* < 0.05 for the indicated experimental points vs the corresponding unstimulated cells, as well as for the unstimulated, R1881- or NGF-stimulated cell pairs, in the absence or presence of y27632. Means and SEMs are shown. *n* indicates the number of experiments. Cells were left untreated or treated as in A and analyzed by IF (**C**) or by WB (**D**) for tubulin β-III. In (**A**,**C**), Bar, 10 μM. In (**E**,**F**), lysates proteins were analyzed by WB, using the antibodies against the indicated proteins.

**Figure 4 cells-12-00373-f004:**
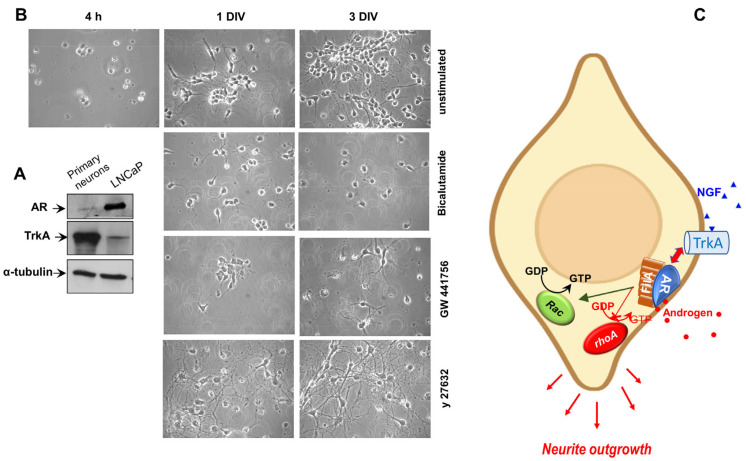
The role of AR, TRKa and ROCK in neuritogenesis. (**A**), Lysates proteins prepared from the indicated cell lines were analyzed by WB, using the antibodies against the indicated proteins. (**B**), Neurons prepared and plated as described in the Materials and Methods section, were unchallenged or challenged as indicated in the figure. Representative contrast phase images of hippocampal neurons at 1 DIV and 3 DIV are represented. (**C**), Model based on our experimental findings. Androgen or NGF induces association of TrkA/AR/FlnA complex, which in turn triggers Rac activation and RhoA inactivation. These events lead to neuritogenesis.

## Data Availability

Not applicable.

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
