# Peer review of "Androgens and NGF Mediate the Neurite-Outgrowth through Inactivation of RhoA"

_cells, 2023, doi:10.3390/cells12030373_

Round 1

Reviewer 1 Report

Data here reported indicate that activation of androgen receptor (AR) or TrkA induces neuritogenesis by inactivation of Rho. Data per se are interesting, however, it is important to highlight that they have been mainly obtained in PC12 and only partially confirmed in hippocampal neurons. In particular, it's a pity that the possible involment of Ser/Thr kinase ROCK has been not evaluated in primary hippocampal neurons. In addition, it is important to note that even if this is an interesting potential mechanism evaluated in vitro, without confirmation in vivo models, its discussion remains very speculative. Indeed, what about a possible role of glial cells? is there a sex dimorphic effect? Indeed, it's unclear the sex of the hippocampal neurons here utilized. In this context see for instance, Ruiz-Palmero I, Sci.Rep. 2016. Additionally, activation of AR is mainly due to dihydrotestosterone and consequently related to the enzyme 5alpha-reductase. Therefore, a discussion on these aspects may render more appealing these results.  

Author Response

First of all, we thank the Reviewer for his/her valuable criticisms that have improved the quality of our manuscript. Below indicated are our answers to each comment. 

  • In the introduction section, on page 2 lines 89-91, we have underlined that the reported findings derive from PC12 cells. Nonetheless, data from the experiments in hippocampal neurons suggest that the formation of the neuronal network involves AR, TrkA and ROCK also in this experimental setting.

  • Concerning the involvement of ROCK, we have argued that an intact catalytic activity is required for the observed effects, as the y27632 compound specifically binds ROCK and impairs its kinase activity by competing with ATP for binding to the catalytic site. Thus, despite the absence of a direct ROCK assay, our findings (Figure 4) show that ROCK inhibition by y27632 enhances the neurite elongation in primary hippocampal cells, which form a typical neuronal network. In this regard, see also our comments on pages 6 (lines 230-231) and 9, lines 336-340  as well as the related reference (Use and properties of ROCK-specific inhibitor Y-27632. Narumiya S, Ishizaki T, Uehata M. Methods Enzymol. 2000;325:273-84. doi: 10.1016/s0076-6879(00)25449-9).

  • Concerning the question “…what about a possible role of glial cells?...”. This is an appropriate comment, as glial cells express androgen and estrogen receptors (Daniel Garci´A-Ovejero et al., The Journal of Comparative Neurology 450:256–271; 2002) and steroid hormones, through these receptors, exert a neuroprotective role after brain damage. We have now added this issue on pages 1 lines 29-32 and 8, pages 280-282 of the revised manuscript. The new reference has been introduced.

  • We completely agree with the Reviewer’s comment that our model should be extended in vivo. Experiments are in progress in our lab using in vivo model of AD. However, they should represent the topic of a different manuscript focused on the role of non-genomic androgen action in neuroprotection and neurodegeneration. Nevertheless, we have highlighted the cons/pros of our present study on page 10, lines 348-356 in the Discussion’s section.

  • Regardless of the sex, we used both male and female embryos and pooled them in our setting (see also the revised version of the manuscript at page 3, lines 115-118 of Methods’s section), as we pointed to underline the role of AR and TrkA in neuritogenesis. In this regard, it should be noticed that, albeit at different extent, female- and male-derived hippocampal neurons express both androgen and estrogen receptors (Isabel Ruiz-Palmero et al., 2017). See also the Result’s section of the revised manuscript on page 7, lines 252-255 and the new Ref. (44).

  • Concerning the comment ‘….activation of AR is mainly due to dihydrotestosterone and consequently related to the enzyme 5alpha-reductase…...’. In the revised version of our manuscript, we have discussed the putative role of 5alpha-reductase in CNS and the findings now reported. In this regard, see the Introduction’s section on page 2, lines 55-57 as well as the Discussion’s section on page 9, lines 301-307 and the new related Refs.  

Reviewer 2 Report

This is an interesting paper. However, a concern is that he authors claim that they "report for the first time that androgens or NGF induce neuritogenesis in PC12 cells etc:" However, there are about ten published papers or more on the effects of androgen on PC12 cells. The authors can download these papers by searching "androgen effects on PC12 cells" . It would be helpful/important for the authors to compare and contrast their findings with those of earlier papers that appear to have a similar aim.

Author Response

  • We thank the Reviewer for the encouraging and appropriate comments.

In the revised manuscript, we have avoided, where possible, the speculation that ‘androgens or NGF induce neuritogenesis in PC12 cells….’. Additionally, some experimental findings from the papers indicated by the Referee have been already presented.

However, many of these papers refer to ectopic AR overexpression in PC12 cells (doi: 10.1016/j.neuroscience.2022.06.034. 10.1111/j.1471-4159.2005.03318.x; 10.1006/hbeh.1994.1035), while others refer to membrane AR or the splicing AR 45 variant (10.1186/s13293-020-0283-1; 10.1016/j.yexcr.2006.04.023). In any case, the mAR has not been cloned so far, to our knowledge and many findings have been observed upon DHEA treatment of PC12 cells (10.1210/en.2007-0645; 10.1096/fj.05-5078fje).

The data we now report, together with our previously published findings (Di Donato et al., 2015) have been exclusively obtained upon R1881 (or DHT) stimulation of PC12 cells, in the absence of AR ectopic expression, since these cells harbor low amounts of endogenous AR, as previously demonstrated by immunoblot, immunofluorescence, somatic knockdown and molecular sequencing (Di Donato et al., 2015).

To meet the Referee’s requests, we have extensively introduced these findings in the Introduction’s section on page 2, lines 60-69 and in the Discussion’s section on pages  8 and 9, lines 288-298. The new Refs have been now included.

Round 2

Reviewer 1 Report

The manuscript has been significantly improved. However, I still remain skeptical for the sex difference. Indeed, many reports by Dr. G. Rune could be useful to left open the possibility in the conclusion of a possible sex difference. Thus, in my opinion the finding that male and female express both androgen and estrogen receptors does not mean that the response is similar.

Author Response

We thank the reviewer for the intriguing comments. To meet these concerns, we have now added in the last version of our manuscript (page 10; highlighted in green) a discussion about the importance of sex in neuroscience studies. We are aware that although hippocampal neurons express the same levels of aromatase, 5-alpha reductase, AR as well as ER and that they release the same quantities of estrogen and androgen, the plot becomes more complex in vivo models. Sex and gender are certainly crucial to obtain results that can eventually be translated into clinical practice. As suggested by the Reviewer, the studies of Dr G. Rune show that the amount of estradiol is extremely low in hippocampal tissue from male or ovariectomized animals. By contrast, significant levels of estradiol can be detected in hippocampal tissue from females. Thus, the in vivo studies go towards the direction of distinguishing the models by sex. Nevertheless, there are still a lot of discrepancies between in vitro/in vivo data.

We agree with the Reviewer about the importance of sex in neuroscience studies. Nevertheless, we have pooled the primary cells, since while some Grant’s applications have to consider the ‘Sex as a Biological Variable’ (Shansky & Woolley, 2016; McCarthy, Woolley, & Arnold, 2017), this procedure has not still been started in Italy. We greatly appreciate the Referee’s suggestions and are conscious that future experiments are needed to dissect the sex-dependency of our findings in in vivo models.

Reviewer 2 Report

Acceptable for publication

Author Response

Thank you